# Impact of Key Nicotinic AChR Subunits on Post-Stroke Pneumococcal Pneumonia

**DOI:** 10.3390/vaccines8020253

**Published:** 2020-05-28

**Authors:** Sandra Jagdmann, Claudia Dames, Daniel Berchtold, Katarzyna Winek, Luis Weitbrecht, Andreas Meisel, Christian Meisel

**Affiliations:** 1Institute for Medical Immunology, Charité-Universitätsmedizin Berlin, Corporate Member of Freie Universität Berlin, Humboldt-Universität zu Berlin, Berlin Institute of Health, 13353 Berlin, Germany; Sandra.jagdmann@charite.de (S.J.); Claudia.dames@charite.de (C.D.); chr.meisel@charite.de (C.M.); 2Department of Experimental Neurology, Charité-Universitätsmedizin Berlin, Corporate Member of Freie Universität Berlin, Humboldt-Universität zu Berlin, Berlin Institute of Health, 10117 Berlin, Germany; daniel.berchtold@charite.de (D.B.); or katarzyna.winek@mail.huji.ac.il (K.W.); Luis.weitbrecht@charite.de (L.W.); 3Center for Stroke Research Berlin, Charité-Universitätsmedizin Berlin, Corporate Member of Freie Universität Berlin, Humboldt-Universität zu Berlin, and Berlin Institute of Health, 10117 Berlin, Germany; 4Neurocure Clinical Research Center, Charité-Universitätsmedizin Berlin, Corporate Member of Freie Universität Berlin, Humboldt-Universität zu Berlin, and Berlin Institute of Health, 10117 Berlin, Germany; 5Department of Neurology, Charité-Universitätsmedizin Berlin, Corporate Member of Freie Universität Berlin, Humboldt-Universität zu Berlin, and Berlin Institute of Health, 10117 Berlin, Germany; 6Labor Berlin, Charité-Universitätsmedizin Berlin Vivantes, 10117 Berlin, Germany

**Keywords:** MCAo, immunosuppression, nicotinic acetylcholine receptor, aspiration-induced pneumonia, *Streptococcus pneumoniae*

## Abstract

Pneumonia is the most frequent severe medical complication after stroke. An overactivation of the cholinergic signaling after stroke contributes to immunosuppression and the development of spontaneous pneumonia caused by Gram-negative pathogens. The α7 nicotinic acetylcholine receptor (α7nAChR) has already been identified as an important mediator of the anti-inflammatory pathway after stroke. However, whether the α2, α5 and α9/10 nAChR expressed in the lung also play a role in suppression of pulmonary innate immunity after stroke is unknown. In the present study, we investigate the impact of various nAChRs on aspiration-induced pneumonia after stroke. Therefore, α2, α5, α7 and α9/10 nAChR knockout (KO) mice and wild type (WT) littermates were infected with *Streptococcus pneumoniae* (*S. pneumoniae*) three days after middle cerebral artery occlusion (MCAo). One day after infection pathogen clearance, cellularity in lung and spleen, cytokine secretion in bronchoalveolar lavage (BAL) and alveolar-capillary barrier were investigated. Here, we found that deficiency of various nAChRs does not contribute to an enhanced clearance of a Gram-positive pathogen causing post-stroke pneumonia in mice. In conclusion, these findings suggest that a single nAChR is not sufficient to mediate the impaired pulmonary defense against *S. pneumoniae* after experimental stroke.

## 1. Introduction

Stroke is a leading cause of death worldwide. The outcome depends on the occurrence of complications. Up to 95% of stroke patients experience medical complications in the first three months after stroke. Among these, infection is one of the most frequent, severe complications [1,2,3,4]. Long-lasting immunosuppression due to overactivation of neurohumoral stress pathways, besides other factors such as neurological deficits leading to dysphagia and aspiration, contribute to the high incidence of pneumonia in stroke patients [5,6,7]. We have previously shown in an experimental mouse model, that excessive cholinergic signaling induced by stroke results in impaired innate immune responses in the lung [8]. In this cholinergic anti-inflammatory pathway, acetylcholine (ACh) released by vagal efferents and by non-neuronal cells was shown to impair antibacterial responses in the lung via the α7 nAChR expressed on alveolar epithelial cells (AECs) and macrophages (MΦ), contributing to an increased susceptibility to spontaneous Gram-negative bacterial pneumonia [8,9,10,11,12]. Nicotine as well as the α7 nAChR-specific agonist PNU282987 diminished LPS-induced IL-6 secretion in AECs isolated from WT but not α7 nAChR KO mice in a dose-dependent manner. In contrast, nicotine and PNU282987 dose-dependently reduced LPS-induced IL-6 secretion in MΦ from WT as well as α7 nAChR KO mice. Thus, nicotine suppressed the TLR-induced pro-inflammatory cytokine secretion in the absence of α7 nAChR, suggesting that suppression of pulmonary immune responses after stroke by the cholinergic anti-inflammatory pathway may in part be independent from the α7 nAChR [8]. nAChRs are homomeric or heteromeric combinations of α2-10 and β3-4 subunits. Beyond the α7 nAChR subunit, mRNA expression of α2 nAChR, α5 nAChR, α6 nAChR, α9 nAChR and α10 nAChR was detected in immune cells including mononuclear leukocytes, dendritic cells (DCs), MΦ and T-cells supporting our hypothesis that not only α7 nAChRs are involved in the regulation of immune response after stroke [12,13,14].

Clinical studies have shown that impaired swallowing, aspiration and stroke-induced immunosuppression contribute to the increased incidence of bacterial pneumonia after stroke [7,15]. Microbiological analysis identified especially Gram-negative bacteria such as *Pseudomonas aeruginosa*, *Klebsiella pneumoniae*, *Enterobacter*, *Escherichia coli* and *Acinetobacter* in blood and lung of patients. However, also Gram-positive bacteria such as *S. pneumoniae*, the leading cause of community-acquired pneumonia, are relevant pathogens causing post-stroke pneumonia [16,17,18]. We have previously shown in an experimental mouse model, that nasal infection with 200 colony-forming units (CFU) of *S. pneumoniae* resulted in severe pulmonary infection after stroke, whereas sham operated mice were able to clear bacteria [19]. β-adrenoreceptor blockade by propranolol treatment significantly reduced bacterial burden in the lung suggesting sympathetic hyperactivity contributes to impaired pulmonary defense after experimental stroke [19,20,21].

In the present study, we aimed to investigate the role of various nAChRs expressed in the lung in the impaired antibacterial responses after stroke in an aspiration-induced model of post-stroke pneumococcal pneumonia.

## 2. Materials and Methods

### 2.1. Animals and Housing

Experiments were executed in accordance with the European directive on the protection of animals used for scientific purposes and further applicable legislation, and approved on 31 March 2016 by the relevant authority, Landesamt für Gesundheit und Soziales (LAGeSo), Berlin, Germany (project identification code: G0244/15). Male α2 nAChR KO (MMRRC_030508-UCD B6.129 × 1-Chrna2^tm1 Jbou^/Mmucd; University of California Davis Mutant Mouse Regional Resource Center (MMRRC)) [22], α5 nAChR KO (MMRRC_000421-UNC B6.129S7-Chrna5^tm1 Mdb^/MmNc; University of North Carolina MMRRC) [23], α7 nAChR KO (JAX #003232B6.129 S7-Chrna7^tm1 Bay^/J; The Jackson Laboratory, Bar Harbor) [24], α9/10 nAChR KO (JAX #005696 CBACaJ;129 S-Chrna9^m1 Bedv^/J; The Jackson Laboratory, Bar Harbor and MMRRC_030509-UCD 129 S4-Chrna10^tm1 Bedv^/Mmcd; University of California Davis MMRRC) [25,26] mice and corresponding WT littermates were used for infection experiments. Standard-genotyping using STR-marker and C57BL/6 substrain-specific mutation analysis confirmed that α2 nAChR KO, α5 nAChR KO and α7 nAChR KO strains carry an autosomal C57BL/6JCrl background (GVG genetic monitoring). α9/10 nAChR KO strain was backcrossed for 8 generations to C57BL/6JCrl. Since all mouse strains carry the same genetic background, mixed WT littermates from all strains were used as control groups (WT MCAo and WT naïve). C57BL/6J mice (The Jackson Laboratory, Bar Harbor, ME, USA) were used for nAChR expression analysis in lung and brain. All animals were housed with identical conditions in cages with chip bedding, mouse tunnel and mouse igloo on a 12 h light/dark cycle with ad libitum access to standard food and water. Experiments were performed with 12–20 weeks old mice.

### 2.2. Experimental Model of Stroke

The surgical procedure of MCAo was performed according to the standard operating procedures of the Department of Experimental Neurology, Charité-Universitätsmedizin Berlin [27]. Under general isoflurane anesthesia, a silicon-coated filament (7019PK5Re, Doccol Corp. Redlands, CA, USA) was introduced into the left common carotid artery and advanced to the origin of the middle cerebral artery (MCA) for 60 min. Infarct volume and success of MCAo was verified by hematoxylin staining from fresh-frozen brains. Animals without infarcts were excluded from the study.

### 2.3. Antibiotic Treatment

Spontaneously developing infection after MCAo was prevented by intraperitoneal (i.p.) injection of marbofloxacin (5 g/kg BW, Vétoquinol GmbH, Ravensburg, Germany) one day before and on the day of MCAo.

### 2.4. Bronchoscopy-Guided Application of S. Pneumoniae Three Days after MCAo

*S. pneumoniae* (D39 capsular type 2 *S. pneumoniae*, Rockefeller University, New York, NY, USA) was grown as described elsewhere [19] and diluted in PBS to 2000 CFU/50 µL. In previous experiments, an optimal dose of bacterial load of 200 CFU for intranasal infection in 129S6SvEv mice was established [19]. Since the C57BL/6J mouse strain used in this study is less susceptible to bacterial infection including *S. pneumoniae* D39 as compared to 129S6SvEv mice [28,29], we established 2000 CFU for infection in previous experiments when developing a miniaturized bronchoscopy protocol in mice [30]. Therefore, we used 2000 CFU for all experiments in this study. The same batch of bacteria from Rockefeller University was used for all experiments.

Under anesthesia with midazolam (5.0 mg/kg BW, Roche Pharma AG, Grenzach-Whylen, Germany) and medetomidin (0.5 mg/kg BW, Orion Corporation, Espoo, Finland) the bronchoscope (Polydiagnost, Pfaffenhofen, Germany) was inserted under visual control into the trachea and advanced to the bifurcation. Subsequently, 50 µL of defined pneumococcal suspension was applied in the main bronchi. Afterward, anesthesia was antagonized subcutaneous (s.c.) with flumazenil (0.5 mg/kg BW, Inresa, Freiburg, Germany) and atipamezol (5 mg/kg BW, Orion Corporation, Espoo, Finland) injection [30].

### 2.5. Microbiological Investigation

Bronchoalveolar lavage (BAL) was performed as described elsewhere [31]. Lungs were removed and homogenized in 500 µL PBS. BAL fluid, blood and lung tissue homogenate were serially diluted, plated on Columbia-Agar plates (BD Bioscience, Heidelberg, Germany), incubated at 37 °C for 18 h and bacterial colonies were counted to calculate the CFUs per ml tissue/liquid.

### 2.6. Flow Cytometry

Isolation of lung cells and splenocytes was performed as described elsewhere [8]. Cell phenotyping was performed on LSRII flow cytometer using FACS Diva software (BD Bioscience, Heidelberg, Germany) and Flowjo software 9.6.6 (Tree Star Inc, San Carlos, CA, USA) with the following anti-mouse monoclonal antibodies: CD45 Peridinin-Chlorophyll-protein (PerCP), CD11b Allophycocyanin-cyanine dye 7 (APC-Cy7), NK1.1 phycoerythrin (PE), CD19 Fluorescein (FITC), CD3 APC, CD4 Alexa Fluor 700 (A700), CD8 Pacific-Blue (PB), Gr1PE, CD11bPE-Cy7, F480 APC, Siglec F APC-Cy7, CD11 c PB (Biolegend, San Diego, CA, USA).

### 2.7. Analysis of Cytokines in BAL and Albumin in BAL and Plasma

Macrophage inflammatory protein-1α (MIP-1α), IL-10, keratinocyte chemoattractant (KC) and tumor necrosis factor α (TNFα) concentration in BAL were measured by using a commercially available Milliplex Map Kit (Merk Millipore, Darmstadt, Germany). Albumin level in BAL and plasma were quantified by an enzyme-linked immunosorbent assay (ELISA) (Bethyl Laboratories Inc., Montgomery, AL, USA).

### 2.8. Quantitative Reverse Transcriptase Polymerase Chain Reaction (qRT-PCR)

RNA from naïve lung and brain was extracted in Trizol according to the manufacturer’s protocol (Roth, Karlsruhe, Germany). All samples were subsequently incubated with DNase (Promega, Fichtburg, MA, USA) followed by purification with Phenol-Chloroform. cDNA synthesis was performed using ProtoScript^®^ II Reverse Transcriptase (New England Biolabs, Ipswich, UK) and the expression of nAChRs was quantified using a LightCycler 480 (Roche, Mannheim, Germany) and the LightCycler-FastStart-DNA-Master-SYBR-Green-I-Kit (Roche, Mannheim, Germany) according to the manufacturer’s guidelines. β-actin was used as “housekeeping gene” for normalization. The following primers were used: mChrnaalpha2 (F: TGGATGGGCTGCAGAGAGACAGG, R: GGTCCTCGGCATGGGTGTGC), mChrnaalpha5 (F: ATCAACATCCACCACCGCTC, R: CTTCAACAACCTCGCGGACG), mChrnaalpha7 (F: TCCGTGCCCTTGATAGCACA, R: TCTCCCGGCCTCTTCATGCG), mChrnaalpha9 (F: CGGACGCGGTGCTGAACGTC, R: AGACTCGTCATCGGCCTTGTTGT), mChrnaalpha10 (F: ACCCTCTGGCTGTGGTAGCG, R: GCACTTGGTTCCGTTCATCCATA). The amplification of Chrna 2, Chrna 7, Chrna 9, Chrna 10 and β-actin was performed at 95 °C (5 s), 66 °C (10 s) and 72 °C (15 s) for 45 cycles. Chrna 5 was amplified with the following conditions: 95 °C (5 s), 60 °C (10 s) and 72 °C (15 s) for 45 cycles. Melting curve analysis was performed to exclude the measurement of non-specific products. PCR products were sequenced to verify primer specificity.

### 2.9. Statistics

Statistical analysis was performed using Prism 6.0 Software (GraphPad, San Diego, CA, USA). Nonparametric one-way analysis of variance (ANOVA) with Dunn’s multiple comparison test was used to compare the mean rank of each group with WT naïve group as a control group.

## 3. Results

### 3.1. α2, α5, α9 and α10 nAChR Subunits are Expressed in Lung and Brain of Naïve Mice

We previously demonstrated expression of α7 nAChR in MΦ and AECs [8]. To investigate which additional nAChR subunits are expressed in lungs and brain, α2, α5, α9 and α10 nAChR mRNA expression in whole organ tissue isolated from naïve mice was quantified by qRT-PCR and compared to α7 nAChR mRNA expression. All subunits were found to be expressed in both lung and brain, however with higher levels in brain compared to lung except for the α10 subunit (Figure 1). These data suggest that other nicotinic receptors in addition to α7 nAChR may be involved in cholinergic suppression of pulmonary immune response after stroke.

### 3.2. Role of Various nAChRs in an Aspiration-Induced Post-Stroke Pneumococcal Pneumonia

To investigate the impact of various nAChRs on the clearance of aspiration-induced pneumococcal pneumonia after experimental stroke, a pneumococcal suspension was applied at the tracheal bifurcation from α2, α5, α7, α9/10 KO mice and WT littermates three days after MCAo. Infected naïve WT mice served as controls. Bacterial burden in the lung, BAL and blood was determined one day after infection. Whereas naïve WT mice were able to clear bacteria, MCAo treated mice showed increased bacterial burden in lung and BAL one day after infection (Figure 2A,B), although the effect in BAL was not significant compared to naïve WT mice. In contrast to naïve WT mice and stroked α7 KO mice, several MCAo-treated α2, α5, α9/10 KO mice and WT littermates suffered from bacteremia after bacterial challenge, whereby α9/10 KO mice exhibited significantly increased bacterial burden in blood compared to naïve WT mice (Figure 2C). However, bacterial burden in lung, BAL and blood was not significantly different between all nAChR KO and WT MCAo groups using WT MCAo mice as the reference group (Figure 2A,C). The infarct size was determined by histological staining and did not differ significantly between WT mice and nAChR KO mice (Figure 2D).

### 3.3. α2, α5, α7, α9/10 nAChRs Have No Effect on Immune Cell Recruitment after Stroke

To investigate the underlying mechanisms of impaired clearance of induced pneumococcal lung infection after stroke and the impact of nAChRs on immune cell recruitment, cellularity in the lung and spleen was determined by flow cytometry one day after infection, which was induced on day three after stroke onset. The number of pulmonary interstitial macrophages (IM) was significantly reduced in α7, α9/10 KO mice and WT littermates and non-significantly reduced in α2 and α5 KO mice compared to naïve WT mice one day after infection. In contrast, MCAo surgery had no effect on the number of neutrophils and lymphocytes in the lung (Figure 3A,C,E). Investigation of cellularity in the spleen revealed a significant decrease in the number of macrophages in α9/10 KO and WT MCAo mice, neutrophils in α5 KO and WT MCAo mice and lymphocytes in α5 KO and α9/10 KO mice compared to infected naïve WT mice (Figure 3B,D,F). The Kruskal–Wallis test using WT MCAo mice as the reference group showed that cell counts of leukocytes in lung and spleen were similar in α2 KO, α5 KO, α7 KO, α9/10 KO and MCAo treated WT mice one day after infection.

### 3.4. Effect of α2, α5, α7, α9/10 nAChRs on Alveolar-Capillary Barrier and Cytokine Secretion in BAL after Stroke

To investigate the inflammatory response during pneumococcal pneumonia after stroke, MIP-1α, IL-10, KC and TNFα concentrations in BAL were measured one day after infection. Cytokine levels tended to be lower in naïve WT mice compared to MCAo mice. MIP-1α, KC, IL-10 and TNFα concentrations were only increased in some mice and did not differ significantly between groups. (Figure 4A–D). Regression analysis between cytokine concentrations and CFU in BAL showed positive correlation between MIP-1α, KC and TNFα level and bacterial burden in BAL (KC: *r* = 0.5186, *p* = 0.0004; MIP-1α: *r* = 0.3391, *p* = 0.028; TNFα: *r* = 0.4492, *p* = 0.0028) (Figure 4E).

To investigate changes in permeability of the alveolar-capillary barrier in the model of aspiration-induced post-stroke pneumonia, we measured albumin concentrations in BAL and plasma one day after infection. Since the plasma albumin concentrations fluctuate, the ratio of BAL albumin and plasma albumin was calculated. It has been shown that albumin ratio in healthy mice is up to three [32]. Here, we found that albumin ratio one day after pneumococcal infection was only increased in some mice and did not differ significantly between the study groups (Figure 4F).

## 4. Discussion

The main finding of the present study is that the depletion of single α nAChR subunits has no effect on the course of an aspiration-induced pneumococcal pneumonia after stroke. Experimental stroke results in severe pneumococcal pneumonia after induced aspiration of *S. pneumoniae* three days after stroke onset, which was harmless for naïve mice. α2 KO, α5 KO, α7 KO, α9/10 KO mice do not show significant differences in the clearance of pathogens, recruitment of immune cells in the lung or pro-inflammatory cytokine secretion compared to WT littermates following stroke.

Pneumonia is the most frequent complication of acute stroke and increases acute and long-term mortality. Besides old age, diabetes mellitus and dysphagia, immunosuppression is recognized as an important contributor for the development of spontaneous infection after stroke. Findings in animals and patients have shown that this impaired peripheral cellular immune response after central nervous system (CNS) injury is a result of activation of the hypothalamic-pituitary-adrenal (HPA) axis, the sympathetic nervous system (SNS) and the cholinergic signaling [5,6,33,34,35,36,37].

Overactivation of the cholinergic signaling in response to infection or inflammation-induced tissue damage is also called ‘the cholinergic anti-inflammatory pathway’ and is described as a protective mechanism that controls the inflammatory response [11]. Peripheral inflammation is sensed by vagal afferent fibers, which leads to an activation of vagal efferent fibers resulting in ACh release in the reticuloendothelial system and interacts with various muscarinic receptors (mAChRs) and nAChRs [10,11]. ACh is not only synthesized as a classical neurotransmitter by parasympathetic nerve fibers but also by non-neuronal cells including airway epithelia cells [38]. This excessive ACh release by non-neuronal and neuronal cells following nervus vagus activation suppresses via nAChRs endotoxin-inducible pro-inflammatory cytokine production, such as TNFα, IL-1β, IL-6 and IL-18 [39].

Until now, five mAChRs subtypes (M1–M5) were found expressed by neuronal and non-neuronal cells including epithelial cells, fibroblasts, smooth muscle cells, macrophages, lymphocytes, mast cells and neutrophils. Non-neuronal mAChRs were shown to mediate inflammation and tissue remodeling in the airways [40]. In the CNS, mAChRs have been associated with the cholinergic anti-inflammatory pathway. An anti-inflammatory and anti-oxidant effect in the hippocampus of rats was demonstrated by stimulation of mAChRs with the agonist oxotremorine [41]. Furthermore, adrenocorticotropin treatment of rats diminished hemorrhagic shock by activation of the cholinergic anti-inflammatory pathway via mAChRs in the CNS [42]. Stimulation of the central localized M1 and M2 receptors resulted in a decrease of TNFα level in blood in an endotoxemia model. Interestingly, a blockade of peripheral mAChRs did not enhance TNFα secretion [43]. These data indicate that stimulation of the mAChRs in the CNS elicits—through cholinergic and potentially other neurotransmitter systems—an anti-inflammatory response in the periphery, whereas anti-inflammatory cholinergic effects in the periphery do not appear to be mediated by mAChRs. Thus, the immunomodulatory effects of cholinergic signaling via mAChRs differ considerably from nAChRs such as α7, which directly exert anti-inflammatory responses in the periphery. Nonetheless, whether central mAChRs play a role in mediating stroke-induced immunosuppression remains to be elucidated.

nAChRs are composed of different combinations of α (α2–α10) and β subunits (β2–β4) and found both in the nervous system and in non-neuronal cells. Structural analysis has shown that α10 subunits form functional channels, when they are co-expressed with α9 and that α9 and α7 subunits are able to form homomeric receptors [44]. So far, especially the α7 nAChR was identified as an important mediator of the cholinergic anti-inflammatory pathway. Previous studies have reported that electrical stimulation of the vagus nerve inhibits the macrophage TNFα release from WT mice but not from α7 nAChR KO mice [9]. Furthermore, it was shown that stimulation of nAChRs with nicotine is associated with decreased neutrophils migration by inhibition of adhesion molecule expression both on the endothelial surface and neutrophils, whereas deficiency of α7 leads to a faster recruitment of neutrophils and decreased bacterial burden after i.p. infection with *Escherichia coli* [45,46]. In addition, α7 nAChR has been demonstrated to play an important role in the development of spontaneous infection after experimental stroke. Depletion of the α7 nAChR by using KO mice reduced bacterial burden in BAL significantly compared to WT littermates [8]. However, the role of nAChRs including the α7 nAChR in pulmonary infections caused by Gram-positive bacteria such as *S. pneumoniae* after stroke had not been investigated, so far.

Expression analysis has shown that α5, α7, α9 and α10 are the most frequently expressed subunits in non-neuronal cells including immune cells [12,13,44]. Besides the anti-inflammatory effect of ACh on macrophages, a cholinergic immunosuppressive effect on human DCs was observed. mRNAs encoding the α2, α5, α6, α7, α10 nAChRs and β2 were found in DC isolated from C57BL/6 J mice suggesting that these subunits mediate anti-inflammatory signaling of DC [47,48]. While the α7 and α9 subunits form either homomeric or, in case of α9 together with the α10 subunit, heteromeric nAChR receptors, respectively, the α2 and α5 subunit co-assembles with other alpha (α3–5) and beta subunits (β2, β4) to different heteromeric receptors [44]. Findings that the α3β2 and α3β4 receptors can also be functional without the α5 subunit suggested that depletion of α5 has no impact on physiological processes and diseases [49]. Nevertheless, α5 KO mice showed reduced hyperalgesia and allodynic responses to carrageenan and complete Freund´s adjuvant (CFA) injections and reduced sensitivity to nicotine-induced seizures and hypolocomotion [23,50,51]. Furthermore, it was shown that the α5 subunit influences the affinity and sensitivity of agonists and antagonists. A 50-fold increased acetylcholine sensitivity was detected if the α5 subunit incorporated with α3β2 [49,52]. Depletion of the α2 subunit resulted in major changes in immune-adipose communication including compromised adaptation to chronic cold challenge, dysregulation of whole-body metabolism and exacerbates diet-induced obesity [53]. In this study, we found expression of α2, α5, α7, α9 and α10 subunits in lungs of naïve mice and suspected that depletion of these subunits may modify the immunomodulatory effects of cholinergic signaling after stroke.

In our experimental stroke model, we have previously shown that signs of immunosuppression in blood, spleen and thymus started as early as 12 h after MCAo with a maximum at day three. Already after five to seven days, lymphocyte numbers and pro-inflammatory cytokines in blood started to recover. Correspondingly, spontaneous bacterial infections in mice were observed between days three and five after experimental stroke [54]. This is in accordance with the clinical observation that the risk to develop infections in stroke patients is highest within two to five days after stroke onset [55]. Therefore, in a translational approach we choose to infect MCAo mice before day five, in the phase of maximum immunosuppression. This time window has been used in aspiration-induced post-stroke pneumonia models using a low number of *S. pneumoniae* capable of inducing severe pneumonia with high bacterial burden in lung of MCAo but not sham animals [19]. Moreover, bacterial burden in lung was lower at day two compared to day one after inoculation [56]. Since naïve mice require usually 24 h to clear induced infection, we used this time point to assess immune parameters and bacterial burden after inoculation of *S. pneumoniae*.

In the present study, we have demonstrated that induced aspiration with *S. pneumoniae* by applying a bacterial suspension at the tracheal bifurcation leads to severe pneumonia after experimental stroke but remains harmless in naïve mice. Microbiological analysis demonstrated significantly lower bacterial burden in WT naive mice compared to stroked animals (Figure 2A). Analysis of the lung showed that out of 15 WT naive mice, 12 mice were able to completely clear inoculated bacteria within 24 h compared to only 3 out of 43 stroked mice (Figure 2A). Microbiological analysis of BAL obtained from the same animals showed similar results (Figure 2B). Moreover, all naive mice remained symptomless in contrast to the MCAo mice suggesting near complete bacterial clearance. This finding corroborates previous findings that stroke impairs the antibacterial defense [7,8,19,20,21,34,54]. *S. pneumoniae* infection in healthy mice leads to neutrophil recruitment in BAL starting 12 h after infection. A reduction of neutrophil numbers was observed 60 h after infection due to cell death [57,58]. Investigation of the lung showed up to 10^6^ neutrophils/mL 24 h after infection, whereas 10^4^ neutrophils/mL were reported in the lung of uninfected mice [59]. In the present study, we could also observe increased numbers of neutrophils in the lung (up to 10^6^ neutrophils/lung) of naïve WT mice and MCAo mice. We found no differences between stroke and naive mice in terms of neutrophil numbers in the lung at day one after infection. One explanation could be a faster kinetic of neutrophil recruitment in naïve animals with already decreasing numbers of neutrophils in the lung one day after bacterial inoculation due to clearance of bacteria. Therefore, a diminished neutrophil recruitment after MCAo would not have been detected with our experimental design. Nevertheless, since we also not observed differences in lung neutrophil counts between WT and nAChR KO mice after stroke, the nAChR subunit status does not seem to have a major impact on neutrophil recruitment into the lung due to *S. pneumoniae* infection after stroke.

Stroke has been demonstrated to induce a long-lasting lymphopenia in blood and spleen starting very early after stroke onset, which is a hallmark of stroke-induced immune depression [54]. In addition, investigations of the lung immunity after stroke have shown a significant reduction of CD4+, CD8+ and B cells 24 h and 72 h after MCAo [60]. Here, we did not observe a significant difference in lung lymphocyte counts between naïve WT and MCAo mice. Investigation of the spleen showed a significant reduction of lymphocyte numbers only in MCAo treated α5 and α9/10 nAChR KO mice compared to naïve WT mice, whereas MCAo treated α2, α7 nAChR KO mice and WT littermates showed no differences compared to naïve WT mice. Notably, we found that infected WT naïve mice also showed diminished lymphocyte counts in the lung compared to normal lymphocyte numbers in untreated WT mice. Clinical data from patients with pneumococcal infections have shown that the acute phase of infection was associated with a diminished number of lymphocytes in blood. Further analysis demonstrated increased apoptosis among lymphocytes [61]. This is in line with experimental data showing increased lymphocyte apoptosis in a mouse model of pneumococcal pneumonia [62]. In addition, *S. pneumoniae* D39 strain has been shown to mediate activation-dependent death in human lymphocytes [63]. Nevertheless, it was reported that the number of CD4+ cells reached normal levels in blood one week after infections suggesting trafficking of CD4+ cells instead of inflow of de novo-generated cells after apoptosis-induced lymphopenia. Therefore, it was assumed that the reduced number of lymphocytes in blood is also caused by migration of lymphocytes to the site of inflammation [61]. This was supported by experiment findings in mice showing that pneumococcal infection increased the number of lymphocytes in BAL compared to uninfected mice [64]. These data suggest that in the present study naïve WT mice developed lymphopenia in spleen and lung due to pneumococcal infection, and consequently, the number of lymphocytes differs only marginally between MCAo mice and naïve WT mice.

Besides alveolar macrophages, a smaller subset of interstitial macrophages (IMs) is found in the lung. While generally IMs are believed to have homeostatic and immunomodulatory functions, these cells may also play an important role in host pathogen defense. Experiments in mice have shown that pulmonary infection induces accumulation of IMs in the lung. In addition, depletion of IMs results in increased bacterial burden after infection suggesting that these cells are essential for controlling pathogens in the lung [65]. Here, we observed a diminished number of IMs in MCAo mice compared to naïve WT mice. These data suggest that the decreased number of IMs in the lung in MCAo mice may contribute to impaired clearance of *S. pneumoniae*.

It is well known that several pneumococcal factors such as opaque variants might disrupt epithelial barriers resulting in a transition of bacteria from the mucosal surface to the bloodstream [66]. Findings in animals and patients have demonstrated that alveolar-capillary barrier disruption leads to increased albumin concentrations in BAL and is associated with neutrophil recruitment into the lung [67,68,69]. Experiments in mice have shown that infection with a lethal dose of *S. pneumoniae* D39 leads to increased albumin level in BAL due to the disruption of the alveolar barrier [70,71]. In healthy mice, the BAL/plasma albumin ratio is 1–3 [32]. Here, we found an increased albumin ratio in several mice, but no differences between groups of various nAChR KO mice, WT littermates and naïve WT mice. Possibly the low impact of pneumococcal infection on the permeability of the alveolar-capillary barrier can be explained by the use of a low infection dose in our model compared to the lethal infection dose in the acute pneumococcal pneumonia mouse model. Neutrophils have been demonstrated to mediate increased epithelial permeability in the lung during pulmonary infection [71,72]. Since nAChR depletion has no effect on neutrophil recruitment into the lung during pneumococcal infection, nAChRs also do not influence the alveolar-capillary barrier 24 h after infection.

In addition, we have investigated the level of pro-inflammatory cytokines in BAL fluid. WT mice without MCAo surgery were able to clear the induced infection within 24 h and did not show elevated levels of pro-inflammatory cytokines in BAL fluid. In contrast, persistent bacterial infection in MCAo mice leads to continuous cytokine secretion, which correlated with bacterial burden in BAL fluid. Experiments in mice have shown that MIP-1α, KC and TNFα level in BAL fluid dramatically increase in the acute phase of pneumococcal pneumonia. [57,59]. In the present study, we could observe that only several MCAo mice were able to induce the pro-inflammatory cytokine response in BAL during the acute phase of pneumococcal infection after stroke. Previous studies have shown that TNFα, KC and MIP-1α secretion is regulated by the cholinergic anti-inflammatory pathway [39,73,74]. These data indicate that stroke impairs pro-inflammatory cytokine secretion during pneumococcal infection contributing to impaired pathogen clearance.

Nevertheless, we found no significant differences between nAChR KO mice and WT mice concerning bacterial burden, cellularity of the lung and spleen, permeability of the alveolar-capillary barrier and cytokine secretion in BAL fluid. These results suggest that various nAChRs including the α7 nAChR do not play a role in increased susceptibility to pneumococcal lung infection after stroke. The apparent discrepancy between earlier studies [8,9,45,46] and the current study concerning the anti-inflammatory effect mediated by the α7 nAChR may be caused by the usage of different types of bacteria in these models. So far, the protective effect of α7 nAChR was solely detected in conjunction with Gram-negative infections. In a mouse model of spontaneous infections after stroke, it was shown that >95% of bacteria cultures from peripheral blood and lung were *E. coli* and depletion of α7 nAChR resulted in diminished bacterial burden in BAL [8,54]. Experiments in mice have shown that the α7 nAChR also mediates impaired immunity in an induced infection with Gram-negative *Pseudomonas aeruginosa* after stroke. α7 nAChR depletion attenuated the effect of stroke on lung injury due to *P. aeruginosa* infection. In contrast, blockade of β-adrenergic receptors by propranolol increased lung injury [75]. Interestingly, propranolol treatment of MCAo mice prevented pulmonary infection with Gram-positive bacteria, such as *S. pneumoniae* [19] and *Listeria monocytogenes* [76]. Many studies investigated the cholinergic anti-inflammatory pathway in the context of uncontrolled inflammatory host response such as sepsis. Findings in animals have demonstrated that stimulation of the cholinergic anti-inflammatory pathway reduced inflammation and mortality in sepsis-induced lung injury due to pulmonary *E. coli* infection and enhanced survival in oral *Salmonella typhimurium* infection [77,78]. Interestingly, nonselective stimulation of nAChRs had no effect on lung inflammation in induced pneumococcal pneumonia with bacteremia [79]. These data suggest that cholinergic activation due to CNS injury may preferentially impair immunity against Gram-negative bacteria. However, further comparative studies including infection models with Gram-negative bacteria and other Gram-positive bacterial strains after stroke are required to further elucidate the immunomodulatory role in cholinergic signaling during infections with different types of bacteria.

## 5. Conclusions

In summary, our findings show that stroke results in impaired pulmonary immunity against *S. pneumoniae* resulting in reduced bacterial clearance and prolonged infection. Blocking of cholinergic signaling by depletion of various nAChRs subunits does not enhance antibacterial immune response suggesting that cholinergic pathways, at least not mediated by α2 nAChR, α5 nAChR, α7 nAChR, α9/10 nAChR subunits, does not play a role in impaired immunity against *S. pneumonia* after stroke. In this respect, it would be interesting to study the impact of non-α7 nAChRs on spontaneous infections or induced-aspiration infections with Gram-negative bacteria, such as *Pseudomonas aeruginosa*, *Klebsiella pneumoniae*, *Enterobacter*, *Escherichia Coli* after stroke.

## Figures and Tables

**Figure 1 vaccines-08-00253-f001:**
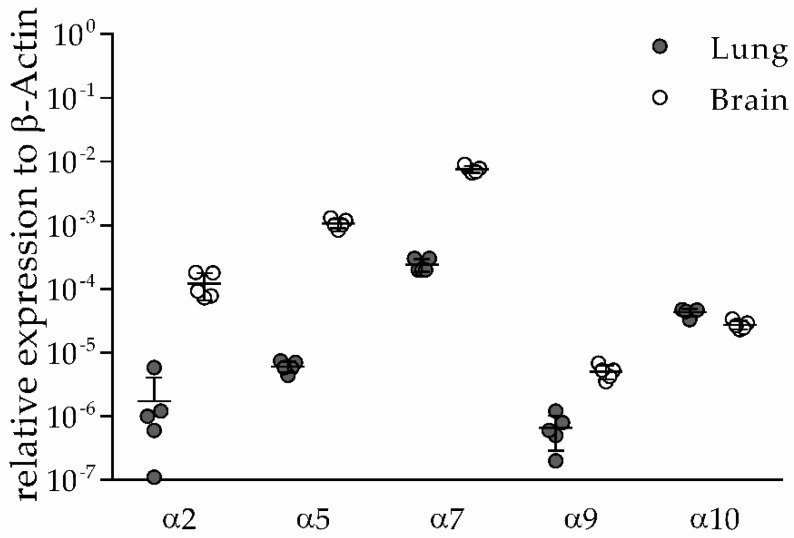
Expression of nAChR subunits in lung and brain tissue. α2, α5, α7, α9 and α10 nAChR subunits are expressed in lung and brain tissue of naïve wild type (WT) mice suggesting a possible role in anti-inflammatory cholinergic signaling after stroke. RNA was isolated from lung and brain tissue, and expression levels were determined by qRT-PCR. Target gene expression was normalized to β-actin as the housekeeping gene. Values are given as mean ± SD (*n* = 5).

**Figure 2 vaccines-08-00253-f002:**
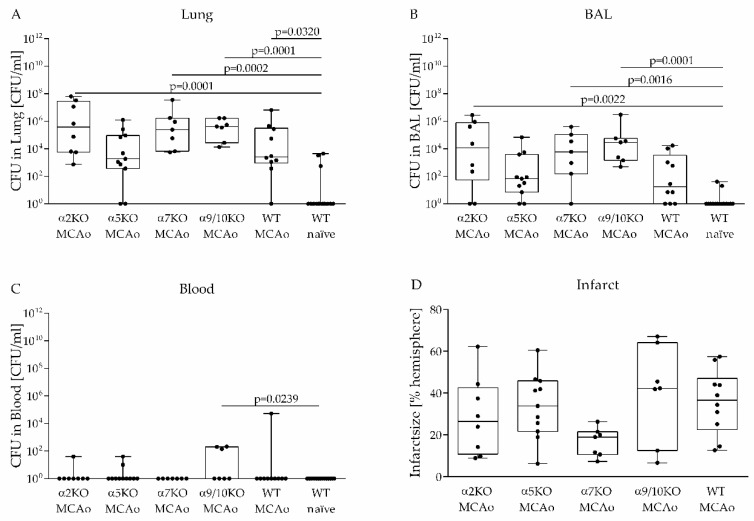
The susceptibility to aspiration-induced pneumococcal pneumonia after experimental stroke is not altered in nAChR knockout (KO) mice. (**A**–**C**) Untreated WT mice (naïve) or WT and nAChR KO mice subjected to MCAo surgery were infected with *S. pneumoniae* three days after MCAo. Microbiological analysis of lung, bronchoalveolar lavage (BAL) and blood was performed one day after infection. Deficiency of α2, α5, α7 and α9/10 nAChRs had no effect on bacterial burden in lung, BAL and blood after experimental stroke. (**D**) nAChRs does not have an impact on infarct size assessed four days after MCAo by histological staining. Data from 6 independent experiments are shown (*n* = 7–15 per group) as box plots compared to WT naïve mice as a reference group for bacterial analysis and compared to WT MCAo mice as the reference group for infarct analysis using the Kruskal–Wallis test followed by Dunn´s test for multiple comparisons.

**Figure 3 vaccines-08-00253-f003:**
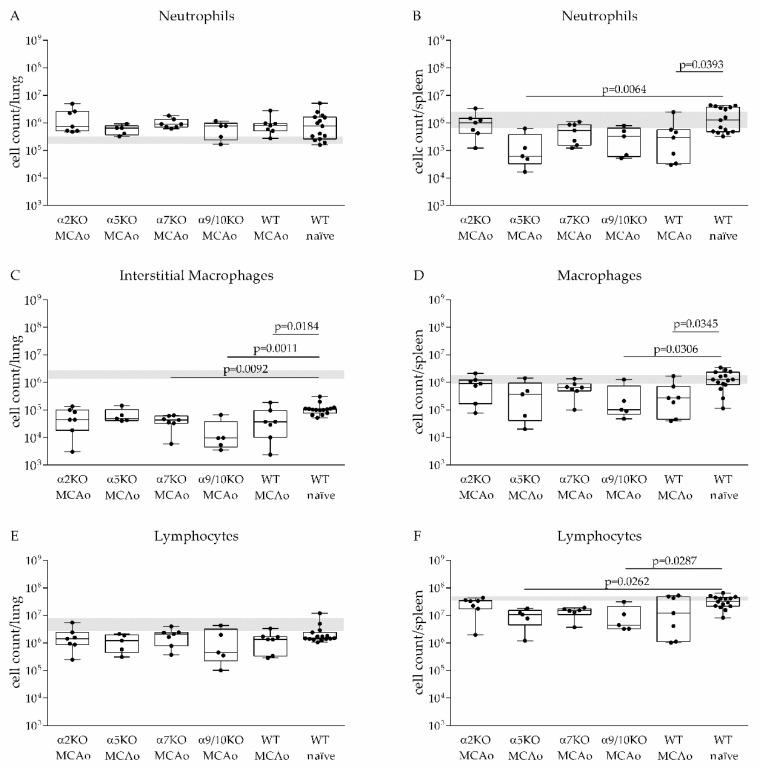
Impact of *S. pneumoniae* infection on cellularity in the lung and spleen of WT and nAChR KO mice. Lung and spleen cells were isolated and quantified by flow cytometry one day after infection with *S. pneumoniae*. WT mice without MCAo surgery show significantly increased numbers of interstitial macrophages (IMs) (CD45/Gr1−/SiglecF−/CD11 b^high^/F480) in the lung (**C**) and increased number of neutrophils (CD45+/CD11 b^high^/Gr1^high^), macrophages (CD45+/Gr1−/CD11 b+/CD11c−) and lymphocytes (B cells: CD45+/CD11 b−/CD19+; T cells: CD45+/CD11 b−/CD3+; NK cells: CD45+/NK1.1+/CD3−; NKT cells: CD45+/NK1.1+/CD3+) in the spleen (**B**,**D**,**F**) compared to MCAo mice. No differences between MCAo mice and naïve WT mice in the number of pulmonary lymphocytes and neutrophils were found (**A**,**E**). Cellularity of the lung and spleen does not differ between WT MCAo mice and nAChR KO MCAo mice. The grey area represents numbers of leukocyte subsets in healthy mice (median with IQR). Data from 6 independent experiments are shown (*n* = 5–15 per group) as box plots compared to naïve WT mice as a reference group using the Kruskal–Wallis test followed by Dunn´s test for multiple comparison.

**Figure 4 vaccines-08-00253-f004:**
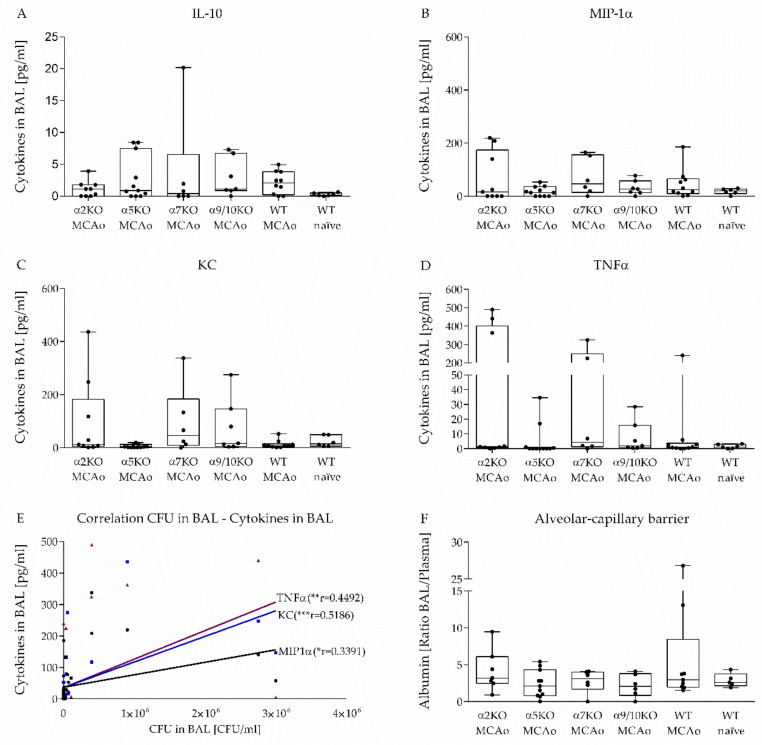
nAChR deficiency has no impact on the alveolar-capillary barrier or cytokine response in lung during *S. pneumoniae* infection after experimental stroke. MIP-1α, IL-10, KC and TNFα concentrations in BAL were measured one day after infection as described in Material and Methods. MIP-1α, IL-10, KC and TNFα level did not differ between groups (**A**–**D**). Positive correlation between MIP-1α, KC and TNFα concentrations and bacterial burden in BAL was found (Pearson Correlation, two tailed) (**E**). To investigate the effect of α2, α5, α7 and α9/10 nAChRs on the permeability of the alveolar-capillary barrier after MCAo, albumin concentrations in BAL and plasma were measured and the ratio of BAL albumin and plasma albumin was calculated. Albumin ratio is increased in several mice but does not differ between nAChR KO mice and WT mice (**F**). Data from 6 independent experiments are shown (*n* = 5–11 per group) as box plots compared to naïve WT mice as a reference group using the Kruskal–Wallis test followed by Dunn´s test for multiple comparison.

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
