# Peer review of "Impact of Key Nicotinic AChR Subunits on Post-Stroke Pneumococcal Pneumonia"

_vaccines, 2020, doi:10.3390/vaccines8020253_

Round 1

Reviewer 1 Report

The authors addressed the questions of whether and which nAChR(s) is involved in the immune system control at the stroke-associated pneumoenia (SAP), focusing on the infection by S. pneumoniae.  In earlier studies, involvement of nAChR alpha 7 was reported in the anti-inflammatory effect to lung infection by Gram-negative bacteria after stroke.  In the manuscript, they present striking features in the case of S. pneumoniae, Gram-positive bacteria.  Their experiments are mostly rationally designed, and data analysis and interpretations are reasonable.  This subject is of general interest to researchers in this field.

To further improve the visibility of the manuscript, I would recommend adding the following descriptions or considerations in the text.

  1. Besides nAChRs, the possible involvement of muscarinic acetylcholine receptors (mAChRs) in the cholinergic anti-inflammatory pathway at the lung infection and SAP. Some such description is expected in Introduction and/or Discussion.
  2. Sampling time windows and its reasonability are not clearly stated. They measured the cellularity in lung and spleen and the cytokine levels in BAL one day after infection.  These parameters may be affected by the time-dependent severity of infection or the magnitude of focal inflammation.

Author Response

Point 1: Besides nAChRs, the possible involvement of muscarinic acetylcholine receptors (mAChRs) in the cholinergic anti-inflammatory pathway at the lung infection and SAP. Some such description is expected in Introduction and/or Discussion.

Response 1: We thank the reviewer for his/her positive feedback on our manuscript. We have added the following paragraph in the discussion dealing with the role of mAChRs in the anti-inflammatory pathway (line 293-309):

Until now, five mAChRs subtypes (M1-M5) were found expressed by neuronal and non-neuronal cells including epithelial cells, fibroblasts, smooth muscle cells, macrophages, lymphocytes, mast cells and neutrophils. Non-neuronal mAChRs were shown to mediate inflammation and tissue remodeling in the airways [1]. In the CNS, mAChRs have been associated with the cholinergic anti-inflammatory pathway. An anti-inflammatory and anti-oxidant effect in the hippocampus of rats was demonstrated by stimulation of mAChRs with the agonist oxotremorine [2]. Furthermore, adrenocorticotropin treatment of rats diminished hemorrhagic shock by activation of the cholinergic anti-inflammatory pathway via mAChRs in the CNS [3]. Stimulation of the central localized M1 and M2 receptors resulted in a decrease of TNFα level in blood in an endotoxemia model. Interestingly, blockade of peripheral mAChRs did not enhance TNFα secretion [4]. These data indicate that stimulation of the mAChRs in the CNS elicits - through cholinergic and potentially other neurotransmitter systems - an anti-inflammatory response in the periphery, whereas anti-inflammatory cholinergic effects in the periphery do not appear to be mediated by mAChRs. Thus, the immunomodulatory effects of cholinergic signaling via mAChRs differ considerably from nAChRs such as α7, which directly exert anti-inflammatory responses in the periphery. Nonetheless, whether central mAChRs play a role in mediating stroke-induced immunosuppression remains to be elucidated.

Point 2: Sampling time windows and its reasonability are not clearly stated. They measured the cellularity in lung and spleen and the cytokine levels in BAL one day after infection.  These parameters may be affected by the time-dependent severity of infection or the magnitude of focal inflammation.

Response 2: We agree with the reviewer. To justify the choice of the experimental setup with defined time points of MCAo, infection and sampling, we added the following paragraph in the discussion with a detailed description (line 349-361):

In our experimental stroke model, we have previously shown that signs of immunosuppression in blood, spleen and thymus started as early as 12h after MCAo with a maximum at day 3. Already after 5 to 7 days, lymphocyte numbers and pro-inflammatory cytokines in blood started to recover. Correspondingly, spontaneous bacterial infections in mice were observed between day 3 and 5 after experimental stroke [5]. This is in accordance with the clinical observation that the risk to develop infections in stroke patients is highest within 2 to 5 days after stroke onset [6]. Therefore, in a translational approach we choose to infect MCAo mice before day 5, in the phase of maximum immunosuppression. This time window has been used in aspiration-induced post-stroke pneumonia models using a low number of S. pneumoniae capable to induce severe pneumonia with high bacterial burden in lung of MCAo but not sham animals [7]. Moreover, bacterial burden in lung was lower at day 2 compared to day 1 after inoculation [8]. Since naïve mice require usually 24h hours to clear induced infection, we used this time point to assess immune parameters and bacterial burden after inoculation of S. pneumoniae.

Reviewer 2 Report

This is an interesting paper in concept, but the results are unexpectedly negative.  The authors had previously published a paper (Stroke. 46:3232-3240, 2015) showing that after a middle cerebral artery occlusion (MCAO) given to mimic a stroke, mice tend to get pneumonia.  Further this effect is significantly blocked by vagotomy or in alpha7 deficient mice suggesting that the cholinergic anti-inflammatory pathway is a reasonable target for therapeutic intervention.  This manuscript is a follow-up study in which the authors infiltrated into the lungs of mice a fixed amount of Streptococcus pneumoniae bacteria after the simulated stroke and tested the effects on wild type mice or on mice with various nicotinic receptor subunit knockouts.  Unexpectedly, they found that nicotinic receptor knockout animals (including alpha7 knockouts) were equally susceptible as the wild-type animals to pneumonia caused by post-MCAO administration of the bacteria.

One problem that the authors did not address is the dose of the bacterial load given after MCAO.  In the earlier Stroke study, the bacterial infections arose spontaneously, and the authors did not identify the bacterial species involved.  In this study, “50 μl of defined pneumococcal suspension (D39 capsular type 2 S. pneumoniae, Rockefeller University, New York) was applied in the main bronchi”. This appears to be such a large dose of this Gram-positive bacteria that naïve animals have a hard time clearing the bacterial load.  In figure 2A, the colony forming units in the lung 1 day after administering the bacteria are barely significantly different in naïve WT than in WT MCAO animals (P=0.03) and are not significantly different in bronchoalveolar lavage (BAL) fluid (Figure 2B).   The authors do not discuss the rationale for using the dose they did and apparently did not test any other doses.  Also, it may be hard for others to replicate their results since they don’t give the method for finding the data in Figure 2A (They do give a reference for determining CFU in BAL for Figure 2B).  Further, the hypothesis is that different batches of bacteria from Rockefeller University are going to cause equivalent infections/volume.  At the very least the authors should justify both the dose given and the timing relative to MCAO and relative to sampling for bacterial load.  Also, what is the basis for the statement in the discussion (lines 306 & 307) “In the present study, we have demonstrated that induced aspiration with S. pneumoniae by applying a bacterial suspension at the tracheal bifurcation leads to severe pneumonia after experimental stroke but remains harmless in naïve mice.” The data presented in this manuscript do not support this.

Further statements in the discussion are not well supported.  In line 309 the authors state “S. pneumoniae infection in healthy mice leads to neutrophil recruitment in BAL starting 12h after infection.” but then state in line 209 “MCAo surgery [in this study] had no effect on the number of neutrophils and lymphocytes in the lung.” (Figure 3A & E). How can the authors expect nicotinic receptor knockouts to rescue a phenomenon (Decreased neutrophil recruitment following MCAO) that does not exist?  It would make more sense for the authors to find a dose of S. pneumoniae that does show a significant decrease in neutrophil recruitment in WT MCAO relative to naïve WT mice and then see if knocking out nicotinic receptors can make a difference.  Similarly, the authors state in line 318 “Stroke has been demonstrated to induce a long-lasting lymphopenia in blood and spleen starting very early after stroke onset, which is a hallmark of stroke-induced immune depression.”  But they did not see any lymphopenia (Figure 3E) in lung following MCAO WT vs. naïve. It is entirely possible that giving a whopping dose of bacteria to even an immune system depressed after stroke will elicit near normal neutrophil and lymphocyte responses.  The same issues apply to the cytokine studies in Figure 4.

Finally, the authors conclude with a discussion that nicotinic receptors may have a role in dealing with Gram-negative bacteria, which seems premature at this point.

The authors did establish that lung tissue expresses mRNA for various alpha nicotinic receptor subunits (a2, a5, a7, a9 & a10), but without beta subunits the a2 & a5 subunits are probably not that important (Figure 1). 

The manuscript is well written, but there are few minor typos: e.g. Line 121, Orion Cooperation should be Corporation.

Author Response

Point 1: One problem that the authors did not address is the dose of the bacterial load given after MCAO.  In the earlier Stroke study, the bacterial infections arose spontaneously, and the authors did not identify the bacterial species involved.  In this study, “50 μl of defined pneumococcal suspension (D39 capsular type 2 S. pneumoniae, Rockefeller University, New York) was applied in the main bronchi”. This appears to be such a large dose of this Gram-positive bacteria that naïve animals have a hard time clearing the bacterial load.  In figure 2A, the colony forming units in the lung 1 day after administering the bacteria are barely significantly different in naïve WT than in WT MCAO animals (P=0.03) and are not significantly different in bronchoalveolar lavage (BAL) fluid (Figure 2B).   The authors do not discuss the rationale for using the dose they did and apparently did not test any other doses.  Also, it may be hard for others to replicate their results since they don’t give the method for finding the data in Figure 2A (They do give a reference for determining CFU in BAL for Figure 2B).  Further, the hypothesis is that different batches of bacteria from Rockefeller University are going to cause equivalent infections/volume.  At the very least the authors should justify both the dose given and the timing relative to MCAO and relative to sampling for bacterial load.  Also, what is the basis for the statement in the discussion (lines 306 & 307) “In the present study, we have demonstrated that induced aspiration with S. pneumoniae by applying a bacterial suspension at the tracheal bifurcation leads to severe pneumonia after experimental stroke but remains harmless in naïve mice.” The data presented in this manuscript do not support this.

Response 1: We agree with the reviewer’s point that the information about the dose of the bacterial load given after MCAo is missing and the reason for choosing that dose was insufficiently described. Therefore, we added the following information to the methods section of our revised manuscript (line 115-123):

  1. pneumoniae (D39 capsular type 2 S. pneumoniae, Rockefeller University, New York) was grown as described elsewhere [7] and diluted in PBS to 2000 CFU/50µl. In previous experiments, an optimal dose of bacterial load of 200 CFU for intranasal infection in 129S6SvEv mice was established [7]. Since the C57BL/6J mouse strain used in this study is less susceptible to bacterial infection including S. pneumoniae D39 as compared to 129S6SvEv mice [9,10], we established 2000 CFU for infection in previous experiments when developing a miniaturized bronchoscopy protocol in mice [11]. The majority of naïve mice were able to clear bacteria while MCAo mice developed severe pulmonary infections. Therefore, we used 2000 CFU for all experiments in this study. The same batch of bacteria from Rockefeller University was used for all experiments.

To enable a replication of our results in figure 2A, we included the method for CFU analysis in the lung as follows (line 133-134):

Lungs were removed and homogenized in 500µl PBS.

To justify the choice of the experimental setup with defined time points of MCAo, infection and sampling, we added a paragraph in the discussion with a detailed description (see our response to point 2 of reviewer 1).

A further point of criticism was that the statement in the discussion section “In the present study, we have demonstrated that induced aspiration with S. pneumoniae by applying a bacterial suspension at the tracheal bifurcation leads to severe pneumonia after experimental stroke but remains harmless in naïve mice.” would not be supported by the current data. This notion referred to the data presented in figure 2A demonstrating significantly lower bacterial burden in WT naive mice compared to stroked animals. For clarification, we added the following paragraph in the discussion to justify our statement (line 365-370):

Microbiological analysis demonstrated significantly lower bacterial burden in WT naive mice compared to stroked animals (Figure 2A). Out of fifteen WT naive mice, twelve mice were able to completely clear inoculated bacteria within 24h. Three mice showed only few CFUs in the lung. In contrast to MCAo mice, these three mice did not show increased pro-inflammatory cytokine level in BAL or clinical signs of infection suggesting near complete bacterial clearance.

Point 2: Further statements in the discussion are not well supported.  In line 309 the authors state “S. pneumoniae infection in healthy mice leads to neutrophil recruitment in BAL starting 12h after infection.” but then state in line 209 “MCAo surgery [in this study] had no effect on the number of neutrophils and lymphocytes in the lung.” (Figure 3A & E). How can the authors expect nicotinic receptor knockouts to rescue a phenomenon (Decreased neutrophil recruitment following MCAO) that does not exist?  It would make more sense for the authors to find a dose of S. pneumoniae that does show a significant decrease in neutrophil recruitment in WT MCAO relative to naïve WT mice and then see if knocking out nicotinic receptors can make a difference.  Similarly, the authors state in line 318 “Stroke has been demonstrated to induce a long-lasting lymphopenia in blood and spleen starting very early after stroke onset, which is a hallmark of stroke-induced immune depression.”  But they did not see any lymphopenia (Figure 3E) in lung following MCAO WT vs. naïve. It is entirely possible that giving a whopping dose of bacteria to even an immune system depressed after stroke will elicit near normal neutrophil and lymphocyte responses.  The same issues apply to the cytokine studies in Figure 4.

Response 2: We thank the reviewer for pointing out this apparent contradiction. In order to clarify our argumentation, we have implemented the following short paragraph in the discussion (line 377-384):

We found no differences between stroke and naive mice in terms of neutrophil numbers in the lung at day 1 after infection. One explanation could be a faster kinetic of neutrophil recruitment in naive animals with already decreasing numbers of neutrophils in the lung 1 day after bacterial inoculation due to clearance of bacteria. Therefore, a diminished neutrophil recruitment after MCAo would not have been detected with our experimental design. Nevertheless, since we also not observed differences in lung neutrophil counts between WT and nAChR KO mice after stroke, the nAChR subunit status does not seem to have a major impact on neutrophil recruitment into the lung due to S. pneumoniae infection after stroke.

We agree with the reviewer that pneumonia itself might have affected lymphocytes, either increased or decreased. To support our hypothesis that pneumococcal pneumonia reduced lymphocyte numbers in naïve mice, we added in the figure 3 reference lines representing number of leukocytes (median with IQR) in healthy mice (without MCAo and without infection) and commented the effect in the discussion as follows (line 387-410):

Figure 3. Impact of S. pneumoniae infection on cellularity in the lung and spleen of WT and nAChR KO mice. Lung and spleen cells were isolated and quantified by flow cytometry one day after infection with S. pneumoniae. WT mice without MCAo surgery show significantly increased numbers of IMs (CD45+/Gr1-/SiglecF-/CD11bhigh/F480+) in the lung (C) and increased number of neutrophils (CD45+/CD11bhigh/Gr1high), macrophages (CD45+/Gr1-/CD11b+/CD11c-) and lymphocytes (B cells: CD45+/CD11b-/CD19+; T cells: CD45+/CD11b-/CD3+; NK cells: CD45+/NK1.1+/CD3-; NKT cells: CD45+/NK1.1+/CD3+) in the spleen (B, D, F) compared to MCAo mice. No differences between MCAo mice and naïve WT mice in number of pulmonary lymphocytes and neutrophils were found (A, E).  Cellularity of the lung and spleen does not differ between WT MCAo mice and nAChR KO MCAo mice. The grey area represents numbers of leukocyte subsets in healthy mice (median with IQR). Data from 6 independent experiments are shown (n= 5-15 per group) as box plots compared to naïve WT mice as reference group using Kruskal-Wallis test followed by Dunn´s test for multiple comparison.

Stroke has been demonstrated to induce a long-lasting lymphopenia in blood and spleen starting very early after stroke onset, which is a hallmark of stroke-induced immune depression [5]. In addition, investigations of the lung immunity after stroke have shown a significant reduction of CD4+, CD8+ and B cells 24h and 72h after MCAo [12]. Here, we did not observe a significant difference in lung lymphocyte counts between naïve WT and MCAo mice. Investigation of the spleen showed a significant reduction of lymphocyte numbers only in MCAo treated α5 and α9/10 nAChR KO mice compared to naïve WT mice, whereas MCAo treated α2, α7 nAChR KO mice and WT littermates showed no differences compared to naïve WT mice. Noteworthy, we found that infected WT naïve mice also showed diminished lymphocyte counts in the lung compared to normal lymphocyte numbers in untreated WT mice. Clinical data from patients with pneumococcal infections have shown that the acute phase of infection was associated with a diminished number of lymphocytes in blood. Further analysis demonstrated increased apoptosis among lymphocytes [13]. This is in line with experimental data showing increased lymphocyte apoptosis in a mouse model of pneumococcal pneumonia [14]. In addition, S. pneumoniae D39 strain has been shown to mediate activation-dependent death in human lymphocytes [15]. Nevertheless, it was reported that the number of CD4+ cells reached normal level in blood one week after infections suggesting trafficking of CD4+ cells instead of inflow of de novo-generated cells after apoptosis-induced lymphopenia. Therefore, it was assumed that the reduced number of lymphocytes in blood is also caused by migration of lymphocytes to the site of inflammation [13]. This was supported by experiment findings in mice showing that pneumococcal infection increased the number of lymphocytes in BAL compared to uninfected mice [16]. These data suggest that in the present study naïve WT mice developed lymphopenia in spleen and lung due to pneumococcal infection and consequently the number of lymphocytes differs only marginally between MCAo mice and naïve WT mice.

Point 3: Finally, the authors conclude with a discussion that nicotinic receptors may have a role in dealing with Gram-negative bacteria, which seems premature at this point.

Response 3: We agree with the reviewer and have soften our statement (line 470-474).

These data suggest that cholinergic activation due to CNS injury may preferentially impair immunity against Gram-negative bacteria. However, further comparative studies including infection models with Gram-negative bacteria and other Gram-positive bacterial strains after stroke are required to further elucidate the immunomodulatory role in cholinergic signaling during infections with different types of bacteria.

Point 4: The authors did establish that lung tissue expresses mRNA for various alpha nicotinic receptor subunits (a2, a5, a7, a9 & a10), but without beta subunits the a2 & a5 subunits are probably not that important (Figure 1). 

Response 4:  We thank the reviewer for this comment. The α2 and α5 subunit co-assembles with different beta subunits including β2 and β4. The Expression of β2 and β4 subunit was also observed in alveolar macrophages and bronchial epithelial cells of the lung [17-19]. We also have found expression of the β2 subunit in lung (as well as brain) tissue, which is shown in a variation of figure 1 presented in our manuscript for scrutiny of the reviewer:  

Modified Figure 1 for scrutiny of the reviewer. Expression of nAChR subunits in lung and brain tissue. α2, α5, α7, α9, α10 and β2 nAChR subunits are expressed in lung and brain tissue of naïve WT mice. RNA was isolated from lung and brain tissue, and expression levels were determined by qRT-PCR. Target gene expression was normalized to β-actin as housekeeping gene. Values are given as mean ± SD (n=5).

We have now added the following paragraph to the discussion section providing the rationale for also investigating a2 and a5 deficient mice in our experimental stroke model (line 332-346):

While the a7 and a9 subunits form either homomeric or, in case of a9 together with the a10 subunit, heteromeric nAChR receptors, respectively, the α2 and α5 subunit co-assembles with other alpha (α3-5) and beta subunits (β2, β4) to different heteromeric receptors [20]. Findings that the α3β2 and α3β4 receptors can also be functional without the α5 subunit suggested that depletion of α5 has no impact on physiological processes and diseases [21]. Nevertheless, α5 KO mice showed reduced hyperalgesia and allodynic responses to carrageenan and complete Freund´s adjuvant (CFA) injections and reduced sensitivity to nicotine-induced seizures and hypolocomotion [22-24]. Furthermore, it was shown that the α5 subunit influences the affinity and sensitivity of agonists and antagonists. A 50-fold increased acetylcholine sensitivity was detected if the α5 subunit incorporated with α3β2 [21,25]. Depletion of the α2 subunit resulted in major changes in immune-adipose communication including compromised adaptation to chronic cold challenge, dysregulation of whole-body metabolism and exacerbates diet-induced obesity [26]. In this study, we found expression of α2, α5, α7, α9 and α10 subunits in lungs of naïve mice and suspected that depletion of these subunits may modify the immunomodulatory effects of cholinergic signaling after stroke.

Point 5: There are few minor typos: e.g. Line 121, Orion Cooperation should be Corporation

Response 5: Typos were corrected (line 126 and 131).

Reviewer 3 Report

This is well writing paper. Some suggestions are highlighted or typed in the attached paper.

Author Response

Point 1: The last part of the introduction appears going beyond going beyond an introductory chapter.

Response 1: We thank this reviewer for his positive feedback on our manuscript. We agree with the reviewer’s point that the last part of our introduction is misplaced and have deleted it (line 75-81).

Correction in line 276 and comment in line 287 were adopted in our revised manuscript.

References

  1. Koarai, A.; Ichinose, M. Possible involvement of acetylcholine-mediated inflammation in airway diseases. Allergology International 2018, 67, 460-466, doi:https://doi.org/10.1016/j.alit.2018.02.008.
  2. Frinchi, M.; Nuzzo, D.; Scaduto, P.; Di Carlo, M.; Massenti, M.F.; Belluardo, N.; Mudò, G. Anti-inflammatory and antioxidant effects of muscarinic acetylcholine receptor (mAChR) activation in the rat hippocampus. Scientific Reports 2019, 9, 14233, doi:10.1038/s41598-019-50708-w.
  3. Guarini, S.; Cainazzo, M.M.; Giuliani, D.; Mioni, C.; Altavilla, D.; Marini, H.; Bigiani, A.; Ghiaroni, V.; Passaniti, M.; Leone, S., et al. Adrenocorticotropin reverses hemorrhagic shock in anesthetized rats through the rapid activation of a vagal anti-inflammatory pathway. Cardiovascular Research 2004, 63, 357-365, doi:10.1016/j.cardiores.2004.03.029.
  4. Pavlov, V.A.; Ochani, M.; Gallowitsch-Puerta, M.; Ochani, K.; Huston, J.M.; Czura, C.J.; Al-Abed, Y.; Tracey, K.J. Central muscarinic cholinergic regulation of the systemic inflammatory response during endotoxemia. Proc Natl Acad Sci U S A 2006, 103, 5219-5223, doi:10.1073/pnas.0600506103.
  5. Prass, K.; Meisel, C.; Höflich, C.; Braun, J.; Halle, E.; Wolf, T.; Ruscher, K.; Victorov, I.V.; Priller, J.; Dirnagl, U., et al. Stroke-induced Immunodeficiency Promotes Spontaneous Bacterial Infections and Is Mediated by Sympathetic Activation Reversal by Poststroke T Helper Cell Type 1–like Immunostimulation. J Exp Med 2003, 198, 725, doi:10.1084/jem.20021098.
  6. Kishore, A.K.; Jeans, A.R.; Garau, J.; Bustamante, A.; Kalra, L.; Langhorne, P.; Chamorro, A.; Urra, X.; Katan, M.; Napoli, M.D., et al. Antibiotic treatment for pneumonia complicating stroke: Recommendations from the pneumonia in stroke consensus (PISCES) group. Eur Stroke J 2019, 4, 318-328, doi:10.1177/2396987319851335.
  7. Prass, K.; Braun Johann, S.; Dirnagl, U.; Meisel, C.; Meisel, A. Stroke Propagates Bacterial Aspiration to Pneumonia in a Model of Cerebral Ischemia. Stroke 2006, 37, 2607-2612, doi:10.1161/01.STR.0000240409.68739.2b.
  8. Mracsko, E.; Stegemann-Koniszewski, S.; Na, S.Y.; Dalpke, A.; Bruder, D.; Lasitschka, F.; Veltkamp, R. A Mouse Model of Post-Stroke Pneumonia Induced by Intra-Tracheal Inoculation with Streptococcus pneumoniae. Cerebrovasc Dis 2017, 43, 99-109, doi:10.1159/000452136.
  9. Jeong, D.-G.; Jeong, E.-S.; Seo, J.-H.; Heo, S.-H.; Choi, Y.-K. Difference in Resistance to Streptococcus pneumoniae Infection in Mice. Lab Anim Res 2011, 27, 91-98, doi:10.5625/lar.2011.27.2.91.
  10. Schulte-Herbrüggen, O.; Klehmet, J.; Quarcoo, D.; Meisel, C.; Meisel, A. Mouse strains differ in their susceptibility to poststroke infections. Neuroimmunomodulation 2006, 13, 13-18, doi:10.1159/000092109.
  11. Dames, C.; Akyüz, L.; Reppe, K.; Tabeling, C.; Dietert, K.; Kershaw, O.; Gruber, A.D.; Meisel, C.; Meisel, A.; Witzenrath, M., et al. Miniaturized bronchoscopy enables unilateral investigation, application, and sampling in mice. Am J Respir Cell Mol Biol 2014, 51, 730-737, doi:10.1165/rcmb.2014-0052ma.
  12. Farris, B.Y.; Monaghan, K.L.; Zheng, W.; Amend, C.D.; Hu, H.; Ammer, A.G.; Coad, J.E.; Ren, X.; Wan, E.C.K. Ischemic stroke alters immune cell niche and chemokine profile in mice independent of spontaneous bacterial infection. Immunity, Inflammation and Disease 2019, 7, 326-341, doi:10.1002/iid3.277.
  13. Kemp, K.; Bruunsgaard, H.; Skinhøj, P.; Klarlund Pedersen, B. Pneumococcal infections in humans are associated with increased apoptosis and trafficking of type 1 cytokine-producing T cells. Infection and immunity 2002, 70, 5019-5025, doi:10.1128/iai.70.9.5019-5025.2002.
  14. Schreiber, T.; Swanson, P.E.; Chang, K.C.; Davis, C.C.; Dunne, W.M.; Karl, I.E.; Reinhart, K.; Hotchkiss, R.S. both gram-negative and gram-positive experimental pneumonia induce profound lymphocyte but not respiratory epithelial cell apoptosis. Shock 2006, 26.
  15. Grayson, K.M.; Blevins, L.K.; Oliver, M.B.; Ornelles, D.A.; Swords, W.E.; Alexander-Miller, M.A. Activation-dependent modulation of Streptococcus pneumoniae-mediated death in human lymphocytes. Pathogens and Disease 2017, 75, doi:10.1093/femspd/ftx008.
  16. McKenzie, C.W.; Klonoski, J.M.; Maier, T.; Trujillo, G.; Vitiello, P.F.; Huber, V.C.; Lee, L. Enhanced response to pulmonary Streptococcus pneumoniae infection is associated with primary ciliary dyskinesia in mice lacking Pcdp1 and Spef2. Cilia 2013, 2, 18-18, doi:10.1186/2046-2530-2-18.
  17. Galvis, G.; Lips, K.S.; Kummer, W. Expression of nicotinic acetylcholine receptors on murine alveolar macrophages. J Mol Neurosci 2006, 30, 107-108, doi:10.1385/jmn:30:1:107.
  18. Carlisle, D.L.; Hopkins, T.M.; Gaither-Davis, A.; Silhanek, M.J.; Luketich, J.D.; Christie, N.A.; Siegfried, J.M. Nicotine signals through muscle-type and neuronal nicotinic acetylcholine receptors in both human bronchial epithelial cells and airway fibroblasts. Respir Res 2004, 5, 27-27, doi:10.1186/1465-9921-5-27.
  19. Maus, A.D.J.; Pereira, E.F.R.; Karachunski, P.I.; Horton, R.M.; Navaneetham, D.; Macklin, K.; Cortes, W.S.; Albuquerque, E.X.; Conti-Fine, B.M. Human and Rodent Bronchial Epithelial Cells Express Functional Nicotinic Acetylcholine Receptors. Molecular Pharmacology 1998, 54, 779, doi:10.1124/mol.54.5.779.
  20. Zoli, M.; Pucci, S.; Vilella, A.; Gotti, C. Neuronal and Extraneuronal Nicotinic Acetylcholine Receptors. Curr Neuropharmacol 2018, 16, 338-349, doi:10.2174/1570159X15666170912110450.
  21. Wang, F.; Gerzanich, V.; Wells, G.B.; Anand, R.; Peng, X.; Keyser, K.; Lindstrom, J. Assembly of human neuronal nicotinic receptor alpha5 subunits with alpha3, beta2, and beta4 subunits. J Biol Chem 1996, 271, 17656-17665, doi:10.1074/jbc.271.30.17656.
  22. Salas, R.; Orr-Urtreger, A.; Broide, R.S.; Beaudet, A.; Paylor, R.; De Biasi, M. The Nicotinic Acetylcholine Receptor Subunit α5 Mediates Short-Term Effects of Nicotine in Vivo. Molecular Pharmacology 2003, 63, 1059, doi:10.1124/mol.63.5.1059.
  23. Kedmi, M.; Beaudet, A.L.; Orr-Urtreger, A. Mice lacking neuronal nicotinic acetylcholine receptor beta4-subunit and mice lacking both alpha5- and beta4-subunits are highly resistant to nicotine-induced seizures. Physiol Genomics 2004, 17, 221-229, doi:10.1152/physiolgenomics.00202.2003.
  24. Bagdas, D.; AlSharari, S.D.; Freitas, K.; Tracy, M.; Damaj, M.I. The role of alpha5 nicotinic acetylcholine receptors in mouse models of chronic inflammatory and neuropathic pain. Biochem Pharmacol 2015, 97, 590-600, doi:10.1016/j.bcp.2015.04.013.
  25. Wang, N.; Orr-Urtreger, A.; Chapman, J.; Rabinowitz, R.; Nachman, R.; Korczyn, A.D. Autonomic function in mice lacking alpha5 neuronal nicotinic acetylcholine receptor subunit. J Physiol 2002, 542, 347-354, doi:10.1113/jphysiol.2001.013456.
  26. Jun, H.; Yu, H.; Gong, J.; Jiang, J.; Qiao, X.; Perkey, E.; Kim, D.-i.; Emont, M.P.; Zestos, A.G.; Cho, J.-S., et al. An immune-beige adipocyte communication via nicotinic acetylcholine receptor signaling. Nature Medicine 2018, 24, 814-822, doi:10.1038/s41591-018-0032-8.

Round 2

Reviewer 2 Report

The major problem with the previous version was that the dose of bacteria was not justified and the experimental methods were not clear enough to be replicated.  The authors have mostly addressed these issues.  This paper should add to literature about the Cholinergic Anti-inflammatory Pathway, which in our hands is not a robust phenomenon, but does seem to exist.

Some minor changes would improve the paper. 

Lines 125-126.  This statement: “The majority of naïve mice were able to clear bacteria while MCAo mice developed severe pulmonary infections.” appears in the methods but looks like a result and should be removed, particularly since the data in figure 2B does not completely support this statement.  The authors may have additional clinical observations that they feel supports this statement, but they should discuss the data that they are presenting to the readers and it should not be in the methods section.  Overall, this description of the methods used is much improved and now should be easily replicated by competent researchers.  On a related note, the authors should revise lines 204-205 in the results to “WT MCAo treated mice showed increased bacterial burden compared to WT controls in lung and BAL one day after infection (Figure 2A-B), although the effect in BAL was not significant.”.

Minor point: In Line 405, “Noteworthy” is an adjective without a noun.  The choices here would be the related adverbs “Notably” or “Noteworthily”.

Author Response

Point 1: Lines 125-126.  This statement: “The majority of naïve mice were able to clear bacteria while MCAo mice developed severe pulmonary infections.” appears in the methods but looks like a result and should be removed, particularly since the data in figure 2B does not completely support this statement.  The authors may have additional clinical observations that they feel supports this statement, but they should discuss the data that they are presenting to the readers and it should not be in the methods section.  Overall, this description of the methods used is much improved and now should be easily replicated by competent researchers.  On a related note, the authors should revise lines 204-205 in the results to “WT MCAo treated mice showed increased bacterial burden compared to WT controls in lung and BAL one day after infection (Figure 2A-B), although the effect in BAL was not significant.”

Response 1: We agree with the reviewer’s point that the statement “The majority of naïve mice were able to clear bacteria while MCAo mice developed severe pulmonary infections.” is more appropriately placed in the results rather than the methods section. We have therefore deleted this sentence in the methods section (line 114-116).

In addition, we have revised our statementWT MCAo treated mice showed increased bacterial burden compared to WT controls in lung and BAL one day after infection (Figure 2A-B)” in the results section as follows (line 187-189):

Whereas naïve WT mice were able to clear bacteria, MCAo treated mice showed increased bacterial burden in lung and BAL one day after infection (Figure 2A-B), although the effect in BAL was not significant compared to naïve WT mice.

In order to clarify our argumentation, we have adapted the paragraph in the discussion as follows (line 358-366):

Microbiological analysis demonstrated significantly lower bacterial burden in WT naive mice compared to stroked animals (Figure 2A). Analysis of the lung showed that out of 15 WT naive mice, 12 mice were able to completely clear inoculated bacteria within 24h compared to only 3 out of 43 stroked mice (Figure 2A. Microbiological analysis of BAL obtained from the same animals showed similar results (Figure 2B). Moreover, all naive mice remained symptomless in contrast to the MCAo mice suggesting near complete bacterial clearance.

Point 2: Minor point: In Line 405, “Noteworthy” is an adjective without a noun.  The choices here would be the related adverbs “Notably” or “Noteworthily”.

Response 2: We thank the reviewer for this notion and corrected the sentence accordingly (line 389):

 Notably, we found that infected WT naïve mice also showed diminished lymphocyte counts in the lung compared to normal lymphocyte numbers in untreated WT mice.

Reviewer 3 Report

Kudos. I think it is as good as it could be humanly.

Author Response

(The authors gave the same response as above.)
